# Alterations in Circulating Bile Acids in Metabolic Dysfunction-Associated Steatotic Liver Disease: A Systematic Review and Meta-Analysis

**DOI:** 10.3390/biom13091356

**Published:** 2023-09-06

**Authors:** Jiaming Lai, Ling Luo, Ting Zhou, Xiongcai Feng, Junzhao Ye, Bihui Zhong

**Affiliations:** Department of Gastroenterology of the First Affiliated Hospital, Sun Yat-sen University, No. 58 Zhongshan II Road, Yuexiu District, Guangzhou 510080, China; laijm6@mail2.sysu.edu.cn (J.L.); luol27@mail2.sysu.edu.cn (L.L.); zhouting_blawan@163.com (T.Z.); fengxc3@mail2.sysu.edu.cn (X.F.)

**Keywords:** bile acid, MASLD, biomarker, disease severity, meta-analysis

## Abstract

**Background:** Previous studies have suggested that bile acids (BAs) may participate in the development and/or progression of metabolic dysfunction-associated steatotic liver disease (MASLD). The present study aimed to define whether specific BA molecular species are selectively associated with MASLD development, disease severity, or geographic region. **Methods:** We comprehensively identified all eligible studies reporting circulating BAs in both MASLD patients and healthy controls through 30 July 2023. The pooled results were expressed as the standard mean difference (SMD) and 95% confidence interval (CI). Subgroup, sensitivity, and meta-regression analyses were performed to address heterogeneity. **Results:** Nineteen studies with 154,807 individuals were included. Meta-analysis results showed that total BA levels in MASLD patients were higher than those in healthy controls (SMD = 1.03, 95% CI: 0.63–1.42). When total BAs were divided into unconjugated and conjugated BAs or primary and secondary BAs, the pooled results were consistent with the overall estimates except for secondary BAs. Furthermore, we examined each individual BA and found that 9 of the 15 BAs were increased in MASLD patients, especially ursodeoxycholic acids (UDCA), taurococholic acid (TCA), chenodeoxycholic acids (CDCA), taurochenodeoxycholic acids (TCDCA), and glycocholic acids (GCA). Subgroup analysis revealed that different geographic regions or disease severities led to diverse BA profiles. Notably, TCA, taurodeoxycholic acid (TDCA), taurolithocholic acids (TLCA), and glycolithocholic acids (GLCA) showed a potential ability to differentiate metabolic dysfunction-associated steatohepatitis (MASH) (all *p <* 0.05). **Conclusions:** An altered profile of circulating BAs was shown in MASLD patients, providing potential targets for the diagnosis and treatment of MASLD.

## 1. Introduction

Metabolic dysfunction-associated steatotic liver disease (MASLD), previously known as nonalcoholic fatty liver disease (NAFLD), affects approximately 32.4% of the adult population worldwide and places a large health and economic burden on global society [1]. According to the Markov model prediction, from 2016 to 2030, different regions will suffer from distinct increasing epidemic trends; for example, the prevalence of MASLD will increase from 17.6% to 22.2% in China and from 26.3% to 28.4% in the United States [2]. Although most patients with MASLD exhibit a benign subtype such as steatosis with or without mild inflammation, others have a necroinflammatory form called metabolic dysfunction-associated steatohepatitis (MASH), which manifests as a progressive form of hepatocellular injury with varied extents of fibrosis [3]. However, the complex pathophysiology of MASLD metabolic dysfunction involved in disease progression requires therapies with multiple targets [4].

Bile acids (BAs), synthesized from cholesterol and metabolized in hepatic cells, play an important role in emulsifying and assimilating dietary fat, cholesterol, and fat-soluble vitamins [5]. Notably, BAs have been discovered as key signaling molecules of the receptors farnesoid X receptor (FXR), Takeda G-coupled protein receptor 5 (TGR5; GPBAR1, M-BAR), pregnane X receptor (PXR), sphingosine-1-phosphate receptor 2 (S1PR2), muscarinic receptors M2/3, and vitamin D receptor [6]. FXR activation can regulate lipid and glucose homeostasis, energy metabolism, hepatic inflammation, and cellular stress and promote intestinal BA uptake [6,7]. TGR5 activation is known to decrease the proinflammatory cytokine response in macrophagocytes [8]. Previous studies have revealed that the disruption of intrahepatic BA homeostasis may be one of the major mechanisms of progression to MASH, cirrhosis, and hepatocellular carcinoma (HCC) [9]. Moreover, determining which subtypes of BAs are more closely related to the occurrence and development of MASLD remains to be investigated. Therefore, identifying the BA signature in MASLD patients may provide new directions for diagnostic or therapeutic methods.

To overcome the limitations related to the small sample size and limited statistical power, we performed a systematic review and meta-analysis of all published studies to explore the difference in circulating BA profiles between MASLD patients and healthy controls. Additionally, the distinctions in circulating BA profile alterations between MASLD patients with different liver disease severities or different disease locations have also been investigated.

## 2. Methods

This meta-analysis was reported in line with the Preferred Reporting Items for Systematic Reviews and Meta-Analysis (PRISMA) [10]. It has been registered in the PROSPERO registry (registration number: CRD42022345481).

### 2.1. Search Strategy

A comprehensive computerized search was performed in the frequently used electronic databases: PubMed, Embase, and Web of Science. The following keywords and synonyms were used: “fatty liver”, “steatohepatitis”, “liver steatosis”, “NAFLD”, “MAFLD”, “MASLD”, “NAFL”, “NASH”, “MASH”, “bile acids”, “bile salts”, and “BA”, which are detailed in Appendix A. There was no restriction on publication dates or geographic regions. The retrieval time was from the establishment of each database to 30 July 2023. The reference lists of all retrieved articles were also manually searched.

### 2.2. Selection Criteria

The inclusion criteria for this study were as follows: (1) observational research, including cross-sectional, case-control, and cohort studies; (2) including both patients diagnosed with MASLD and healthy controls; (3) studies reporting either serum/plasma BAs or specific individual BA concentrations or providing accessible data necessary to calculate them; and (4) if more than one published study included the same population, the most recent or most detailed study was included to avoid data duplication. The exclusion criteria were as follows: (1) reviews, editorials, conference abstracts, and case reports; (2) non-English studies; (3) studies on animals or cell lines; (4) patients with MASLD combined with other liver diseases, such as alcoholic fatty liver disease, autoimmune liver disease, and viral hepatitis; and (5) participants who received intervention at baseline.

### 2.3. Data Extraction

Two authors (Lai J. and Luo L.) were responsible for selecting studies and extracting data from the included studies. For any differences in the extractions, the corresponding authors (Ye J and Zhong B) participated in the discussion to reach an agreement. Using a standardized data extraction form, the following information was recorded: first author, year of publication, geographical location, study design, study group, diagnostic methods, sample size, age, sex ratio, body mass index (BMI), and alanine aminotransferase (ALT) of the MASLD patients and controls; detection methods of BAs; and the circulating concentrations of total or each individual BA.

If important and necessary data could not be extracted directly from the articles, we contacted the corresponding or first authors by email to acquire the original data. If they did not reply, the articles were excluded. When the original data were unavailable, we also contacted the authors for assistance. Because the concentration units were different, we uniformly transferred them into μmol/L before pooling the data. In the meta-analysis, the medians with an interquartile range (IQR) for concentration were transferred into the means with a standard deviation (SD). All data transformations were performed by applying standard statistical formulas [11].

### 2.4. Quality Assessment

Study quality was assessed using the Newcastle-Ottawa Scale (NOS) criteria by two investigators (Lai J. and Luo L.). The NOS, which was proposed by Wells and his colleagues, is a scale for evaluating the quality of published nonrandomized studies in meta-analyses. The NOS comprises eight items, categorized into three criteria: selection, comparability, and outcome (cohort studies) or exposure (case-control studies). The quality score ranged from zero to nine and was classified as follows: poor quality (0–3), moderate quality (4–6), and high quality (7–9) [12].

### 2.5. Statistical Analysis

Statistical analysis was performed using Review Manager Version 5.4 and Stata 12.0. After data extraction and transformation, all BA concentration data were presented as means and SDs. Taking into account the significant differences in the means among the included studies, the effect size was expressed as the standardized mean differences (SMDs) with 95% confidence intervals (CIs). Between-study heterogeneity was evaluated by the *I*^2^ statistic with a significance level of >50%, and it is better to choose the random-effects model rather than the fixed-effects meta-analysis model [13].

The potential for publication bias was evaluated by the visual inspection of funnel plot asymmetry, and the bias was further quantified by Begg’s and Egger’s tests, with significance levels set at *p*-value < 0.05 [14]. If publication bias was present, we further used the trim-and-fill method to assess the influence of this bias on the estimates [15]. The Grading of Recommendations Assessment, Development and Evaluation (GRADE) assessment was used to establish the certainty of evidence for each meta-analysis [16]. To identify the sources of heterogeneity, subgroup analysis was first performed based on geographic region (Eastern and Western countries) and disease severity (non-MASH and MASH). Then, random-effects meta-regression analysis was conducted, where the mean age, percentage of males, BMI, and ALT of MASLD patients were regarded as moderators. Meanwhile, sensitivity analysis was performed by examining the effect of any individual study on the estimated effect size of the outcome.

## 3. Results

### 3.1. Study Selection

As shown in Appendix A, a total of 4362 records were initially identified by using the predefined search strategy from PubMed (*n* = 1224), Embase (*n* = 1562), and Web of Science (*n* = 1576). After screening the title and abstract, 992 duplicate articles were deleted, and 3335 irrelevant studies were removed. Of the remaining 35 studies that were assessed for eligibility by reviewing the full text, one analyzed a duplicate cohort, two had no control groups, and thirteen lacked available serum/plasma BA concentration data. Ultimately, 19 studies published from 2013 to 2023 were included for qualitative synthesis in this meta-analysis [17,18,19,20,21,22,23,24,25,26,27,28,29,30,31,32,33,34,35]. After additional checks of the references listed in these nineteen studies, no additional eligible studies were obtained.

### 3.2. Characteristics and Quality Assessment of the Included Studies

Table 1 outlines the basic characteristics of the included studies. A total of 154,807 individuals, including 43,229 MASLD patients and 111,578 healthy controls, were included. Of the nineteen studies, six were performed in China, five in the USA, three in Japan, one in Korea, one in Australia, one in Germany, one in Guatemala, and one each in Italy and Austria. Fourteen studies diagnosed MASLD by liver biopsy, two by ultrasound, two by ultrasonography and/or liver biopsy, and one by fatty liver index (FLI). Sixteen studies were conducted in adult patients with MASLD; two were performed in children with MASLD; and the remaining study was conducted in adults with both type 2 diabetes mellitus and MASLD. Furthermore, we performed a quality assessment of all included studies by the NOS criteria, and the general quality was considered moderate to high (nine studies scored 8 points, seven scored 7 points, and three scored 6 points, mean ± SD: 7.32 ± 0.75) (Appendix A). No article was excluded because of low quality.

Based on the available data we obtained, this meta-analysis focused on alterations at different levels, including total BAs, total unconjugated BAs, total conjugated BAs, total primary BAs, secondary BAs, and 15 individual BAs. Notably, 15 molecular BA species included in the present meta-analysis were cholic acids (CA), deoxycholic acids (DCA), chenodeoxycholic acids (CDCA), ursodeoxycholic acids (UDCA), lithocholic acids (LCA), glycocholic acids (GCA), glycodeoxycholic acids (GDCA), glycochenodeoxycholic acids (GCDCA), glycoursodeoxycholic acids (GUDCA), glycolithocholic acids (GLCA), taurococholic acids (TCA), taurodeoxycholic acids (TDCA), taurochenodeoxycholic acids (TCDCA), tauroursodeoxycholic acids (TUDCA) and taurolithocholic acids (TLCA).

### 3.3. Meta-Analysis of Circulating Bile Acids

As presented in Figure 1, the results of the meta-analysis indicated that there was a significant difference in total circulating BA concentrations between MASLD patients and healthy controls, with a total random-effects SMD of 1.03 (95% CI: 0.63–1.42, *p* < 0.001) and substantial heterogeneity (*I*^2^ = 96%). Furthermore, we divided total BAs into either unconjugated and conjugated BAs or primary and secondary BAs, showing that, except for the secondary BAs (SMD = 0.48, 95% CI: −0.33–1.29, *p* = 0.25), the pooled results were consistent with the overall estimate (Figure 1). Based on available data, this meta-analysis also focused on 15 molecular species of circulating BAs and revealed that 9 of the 15 BAs were increased in MASLD patients (Figure 2 and Figure 3). Of note, the top five individual BAs were UDCA, TCA, CDCA, TCDCA, GCA, GUDCA, GCDCA, TUDCA and CA, with total SMDs of 0.86 (95% CI: 0.44–1.28, *p* < 0.001), 0.78 (95% CI: 0.50–1.06, *p* < 0.001), 0.74 (95% CI: 0.40–1.09, *p* < 0.001), 0.65 (95% CI: 0.35–0.94, *p* < 0.001), 0.65 (95% CI: 0.39–0.91, *p* < 0.001), 0.63 (95% CI: 0.32–0.95, *p* < 0.001), 0.62 (95% CI: 0.35–0.90, *p* < 0.001), 0.52 (95% CI: 0.15–0.90, *p =* 0.006) and 0.47 (95% CI: 0.19–0.75, *p =* 0.001), respectively. However, there were no significant differences in DCA, LCA, GDCA, TDCA, GLCA, or TLCA levels between the MASLD patients and control groups.

### 3.4. Subgroup Analysis

The meta-analysis results of the subgroup analyses are detailed in Table 2 and Table 3. When subgroup analyses were conducted by geographic location, it was found that MASLD patients had higher levels of total circulating BAs than healthy controls in Eastern countries (SMD = 1.40, 95% CI: 0.75–2.05, *p* < 0.001) but not in Western countries (SMD = 0.61, 95% CI: −0.05–1.28, *p* = 0.07) (Table 2). Regarding total unconjugated and conjugated BAs, MASLD patients presented increased concentrations in both the Eastern and Western subgroups. However, in the analysis of total primary and secondary BAs, Eastern countries only showed elevated primary BA levels (SMD = 1.49, 95% CI: 0.12–2.86, *p* = 0.03), while Western countries showed elevated levels of both primary and secondary BAs (SMD = 0.65, 95% CI: 0.12–1.18, *p* = 0.02 for primary BAs, and SMD = 1.06, 95% CI: 0.15–1.98, *p* = 0.02 for secondary BAs). Furthermore, further analysis of individual BAs demonstrated that MASLD patients in Eastern countries had elevated levels of nine molecular species of BAs, including CA, CDCA, UDCA, GCA, GCDCA, GUDCA, TCA, TCDCA, and TUDCA, while those in Western countries exhibited higher levels of ten individual BAs, including DCA, CDCA, UDCA, GCA, GDCA, GCDCA, GUDCA, TCA, TDCA, and TCDCA (all *p* < 0.05). 

Moreover, in the subgroup analysis by disease severity, it was found that non-MASH and MASH exhibited different circulating BA profiles (Table 3). Compared with control populations, a significant increase in total BA levels was found in patients with MASH (SMD = 1.59, 95% CI: 0.53–2.65, *p* = 0.003) but not in those with non-MASH (SMD = 0.04, 95% CI: −1.06–1.15, *p* = 0.94). A stratified analysis revealed that non-MASH patients had elevated primary and unconjugated BA levels, whereas MASH patients had higher circulating levels of not only primary and unconjugated BAs but also secondary and conjugated BAs. Furthermore, non-MASH patients presented increased CDCA, GCA, GUDCA, and TCA concentrations and decreased TLCA levels, with total SMDs of 0.26 (95% CI: 0.05–0.48, *p* = 0.02), 0.51 (95% CI: 0.06–0.96, *p* = 0.03), 0.47 (95% CI: 0.14–0.81, *p* = 0.006), 0.45 (95% CI: 0.03–0.88, *p* = 0.04), and −0.38 (95% CI: −0.70, −0.06, *p* = 0.02), respectively. However, MASH patients had elevated concentrations of almost all BA molecular species except CA and DCA. Notably, the pooled results revealed that total conjugated BA levels were higher in patients with MASH than in those without MASH (SMD = 1.28, 95% CI: 0.94–1.62, *p* < 0.001). More specifically, TLCA, TCA, TDCA, and GLCA were the molecular species of BAs that could distinguish between MASH and non-MASH (SMD = 0.53, 95% CI: 0.02–1.05, *p* = 0.04 for TLCA; SMD = 0.47, 95% CI: 0.24–0.71, *p* < 0.001 for TCA; SMD = 0.40, 95% CI: 0.08–0.71, *p* = 0.01 for TDCA; and SMD = 0.36, 95% CI: 0.003–0.71, *p* = 0.045 for GLCA).

We further performed subgroup analysis based on the detection samples, showing that the results of the serum samples were similar to the overall results (Appendix A). Of note, four of the 15 plasma bile acids did not provide enough data to calculate the pooled results. Additionally, we conducted a subgroup analysis of diagnostic methods and found that the histology subgroup had higher levels of most circulating bile acids, except for LCA and TLCA (Appendix A).

### 3.5. Publication Bias and Sensitivity Analysis

The presence of publication bias in the total circulating BA and 15 individual BAs was assessed by observing asymmetry in the funnel plots and performing Egger’s regression test and Begg’s test. As shown in Appendix A, Egger’s test revealed significant publication bias for total BAs (*p* = 0.017). The trim-and-fill method was further performed to identify and correct the results. However, these meta-analysis results were unchanged. Thus, publication bias did not alter the statistical significance of the estimates, indicating that this bias had a minimal effect on the pooled results.

Furthermore, we performed a sensitivity analysis to investigate whether any studies would alter the significance of the SMDs of total BAs and distinct molecular species of BAs (Appendix A). As a result, it showed no significant change in the results of the analysis for all categories. However, the GRADE quality assessment indicated very low confidence in the overall pooled evidence regarding the pooled result of total BAs, unconjugated and conjugated BAs, primary and secondary BAs, and 15 individual BAs (Appendix A).

### 3.6. Meta-Regression Analysis

When considering available data, we further performed a random-effects meta-regression analysis to investigate the impact of several continuous variables on circulating BAs, including the mean age, male proportion, BMI, and ALT of MASLD patients. The results of the univariate meta-regression are summarized in Table 4, and bubble plots are shown in Appendix A. When analyzing circulating concentrations of CDCA and UDCA, age was a slight but significant confounding factor (regression coefficient beta = −0.045, *p* = 0.03 for CDCA, and regression coefficient beta = −0.034, *p* = 0.04 for UDCA). However, no other significant correlation was found in any other pair of comparisons.

## 4. Discussion

To our knowledge, the current study is the first meta-analysis on this topic and the most comprehensive systemic review to demonstrate an overall profile of circulating BAs in MASLD. The present study of 43,229 MASLD patients and 111,578 healthy controls showed a significant relationship between circulating BAs and MASLD by evaluating the levels of total or distinct molecular species of BAs. More specifically, of the 15 individual BAs extracted in this meta-analysis, 9 were increased in MASLD patients, and the top 5 were UDCA, TCA, CDCA, TCDCA, and GCA, whereas no significant differences in DCA, LCA, GDCA, TDCA, GLCA, or TLCA levels were observed between the two groups.

Under normal conditions, BAs serve as important signaling molecules and exert a critical role in the regulation of nutrient digestion and metabolism. In particular, there was a complex and potentially pathogenic interrelationship between gut microbiota and BAs. Specifically, the gut microbiota could modulate the BA pool size and composition through enzymatic activities [4,36]. On the other hand, BAs were found to shape the intestinal microbiome by exerting antimicrobial effects, stimulating spore germination, and serving as sources of nutrients [37,38]. However, the present meta-analysis provided strong evidence for significant elevations in total BAs and 9 of 15 individual BAs (including UDCA, TCA, CDCA, TCDCA, GCA, GUDCA, GCDCA, TUDCA, and CA) in MASLD patients, indicating that increased BA exposure may be involved in the pathogenesis of MASLD.

One study by Jiao N et al. demonstrated that although both primary and secondary BA levels were indeed increased in MASLD, FXR-mediated and fibroblast growth factor receptor 4 (FGFR4)-mediated signaling was impaired [7,21]. Previous studies by Nimer N. et al. and Kasper M. et al. revealed that the expression of patatin-like phospholipase domain-containing 3 (PNPLA3) was associated with BA metabolism [30,39]. Thus, another explanation was that PNPLA3, which has been linked to multiple lipid metabolic processes and liver fibrosis, bridged the association between BA metabolism and MASLD development. In addition, BA accumulation may predispose individuals to MASLD through direct hepatotoxicity, inducing inflammatory cytokines, gut microbiota imbalances, and hyperendotoxemia caused by increased intestinal permeability [40,41,42].

So far, liver biopsy is still the “gold standard” method for staging steatosis, inflammation, and fibrosis in MASLD, but it cannot be routinely used in clinical practice. Therefore, it is important to seek potential serum biomarkers for the assessment of MASLD. A cohort study found that the plasma TBA levels in MASH patients are significantly higher than those in non-NASH patients, regardless of their T2DM status. Through multiple linear regression analysis, plasma TCA and GCA levels could effectively distinguish MASH, independent of other confounding factors of T2DM [43]. Another study showed that, through logistic regression analysis, increasing plasma TCA increased the likelihood of severe steatosis, elevated GCA levels significantly increased the likelihood of inflammation, and higher plasma TCA was associated with a significantly higher likelihood of ballooning. As determined by least squares regression analyses, higher plasma total primary BAs were associated with a higher NAFLD activity score and an increased likelihood of fibrosis ≥ 2 [22]. In our subgroup analysis, the GCA and TCA of MASH were also higher than those of healthy controls, consistent with the above research findings. In the overall result, all primary bile acids were also consistently elevated, which confirms that BAs can serve as essential biomarkers to distinguish different stages of MASLD without biopsy.

Based on the pooled results, we hypothesized that 9 of the 15 individual BAs may be potential diagnostic markers and therapeutic targets for MASLD, especially UDCA, TCA, CDCA, TCDCA, and GCA, which were elevated to a higher degree in MASLD patients than in healthy individuals. It was in line with several previous reports. Previous observations have demonstrated that UDCA has anti-inflammatory and anti-apoptotic effects [44]. However, another meta-analysis of 8 studies with 655 participants showed that UDCA treatment significantly reduced blood concentrations of alanine aminotransferase (ALT) and γ-glutamyl transferase but did not show any significant effect on anthropometric characteristics or hepatic histology [45]. Additionally, published studies pointed to TCA not only as a biomarker for liver injury but also as a causal factor that disturbs lipid metabolism [46]. Research on HFD-induced obesity rats (also commonly used as a MASLD model) showed that HFD promoted BA intestine passive absorption to increase the concentrations of BAs, especially CDCAs, which activated the Fxr-Fgf15 pathway in the ileum to result in the mRNA expression of Cyp7a1 in liver down-regulation, which inhibited cholesterol metabolizing into primary BAs and contributed to the cholesterol level increase [47]. Another animal experiment suggested that TCDCA produces toxicity in mouse primary hepatic cells and induces mitochondrial permeability transition and Caspase-11 pyroptosis in mice [48]. Serum GCA may be associated with genetic variation in the TRIB1 gene, and functionally, TRIB1 may influence hepatic triglyceride synthesis and secretion in humans [31]. Notably, the relatively small sample size may result in low statistical power, and it was doubted whether the results of animal experiments could be demonstrated on humans, so further human and mechanistic studies need to be carried out to clarify the results.

MASH is generally considered to be progressive, but its pathogenesis has not been fully elucidated [49,50]. Interestingly, pooling all currently available data, our study confirmed that the circulating BA profile varied by disease severity, as characterized by the significant elevation of total conjugated BAs in MASH. Furthermore, when examining individual molecular species of BAs, the concentrations of these four individual BAs, including TLCA, TCA, TDCA, and GLCA, were significantly higher in MASH than in non-MASH. This phenomenon may be attributed to alterations in the gut microbial composition; namely, taurine- and glycine-metabolizing bacteria were obviously increased in the gut of MASH patients [21]. Therefore, we suspected that these four BAs may be key factors for adverse liver remodeling during the clinically relevant transition between non-MASH and MASH. Consistent with this hypothesis, a study investigating liver BA profiles demonstrated that increased concentrations of taurine-conjugated BAs (e.g., TCA and TDCA) were found in MASH livers [51]. The exact mechanisms remain unclear, but experimental evidence has reported that circulating TLCA can impair bile flow and induce cholestasis [52]. Additionally, TCA can activate hepatic stellate cells by upregulating the Toll-like receptor 4 (TLR4) signaling pathway, thereby playing an important role in both liver remodeling and the development of portal hypertension [46,53,54]. Interestingly, our study also revealed low TLCA concentrations in non-MASH but high TLCA levels in MASH. One plausible reason for this contradiction is that metabolic changes (e.g., hyperinsulinemia) may inhibit BA synthesis and reabsorption in early MASLD. However, mechanical factors (e.g., altered blood flow and fibrosis) lead to increased levels of BAs with MASLD progression [19]. Nonetheless, future studies are encouraged to clarify the detailed mechanism and identify the clinical utility of TLCA, TCA, TDCA, and GLCA to monitor disease progression.

Moreover, another important finding in the current study was that different regions exhibited unequal circulating BA profiles. In brief, GCA and TUDCA were the most significantly elevated in Eastern countries, while TCA and GCA were elevated in Western countries. Notably, this result, especially GCA, provided valuable clues for the selection of a novel potential treatment target for MAFLD patients in different geographic locations. This discrepancy may have occurred because the gut microbiota has racial and geographic disparities, leading to varied BA patterns [55].

There were several limitations to this systemic review. First, the number of studies included in the present study was small, and seven studies included fewer than 50 MASLD patients. Notably, the statistical capacity may be low because of the limited studies, suggesting that the pooled results should be interpreted with caution. However, we comprehensively and systematically searched the database to identify most articles. Second, the statistical heterogeneity of MASLD definitions precludes accurate cross-study comparisons. Subgroup and sensitivity analyses were further conducted and revealed that geographic regions, disease severity, MASLD detection methods, and BAs test samples may be confounders. However, other important factors, including varied lifestyles and the degree of steatosis, could not be assessed due to a lack of sufficient data. Third, there was a possible publication bias in the current study. When screening the records, unpublished studies such as dissertations or conference abstracts were excluded. Nonetheless, trim-and-fill analyses indicated that the impact of this bias on our results was likely insignificant, and the sensitivity of the publication bias test was considered low due to the limited number of included studies. Finally, 14 of the 19 included studies were case-control studies, which could not demonstrate the cause-and-effect relationship between BAs and MASLD. Therefore, larger longitudinal cohort studies are warranted to investigate whether circulating BAs contribute to the occurrence and development of MASLD.

## 5. Conclusions

Our meta-analysis demonstrated that excess BA production may be involved in liver injury and the development and/or progression of MASLD, and the circulating BA profile in MASLD patients varied by disease severity and geographic region. This provided an important clue for the search for potential MASLD diagnostic and therapeutic targets. However, these findings warrant further investigation and validation (Appendix A).

## Figures and Tables

**Figure 1 biomolecules-13-01356-f001:**
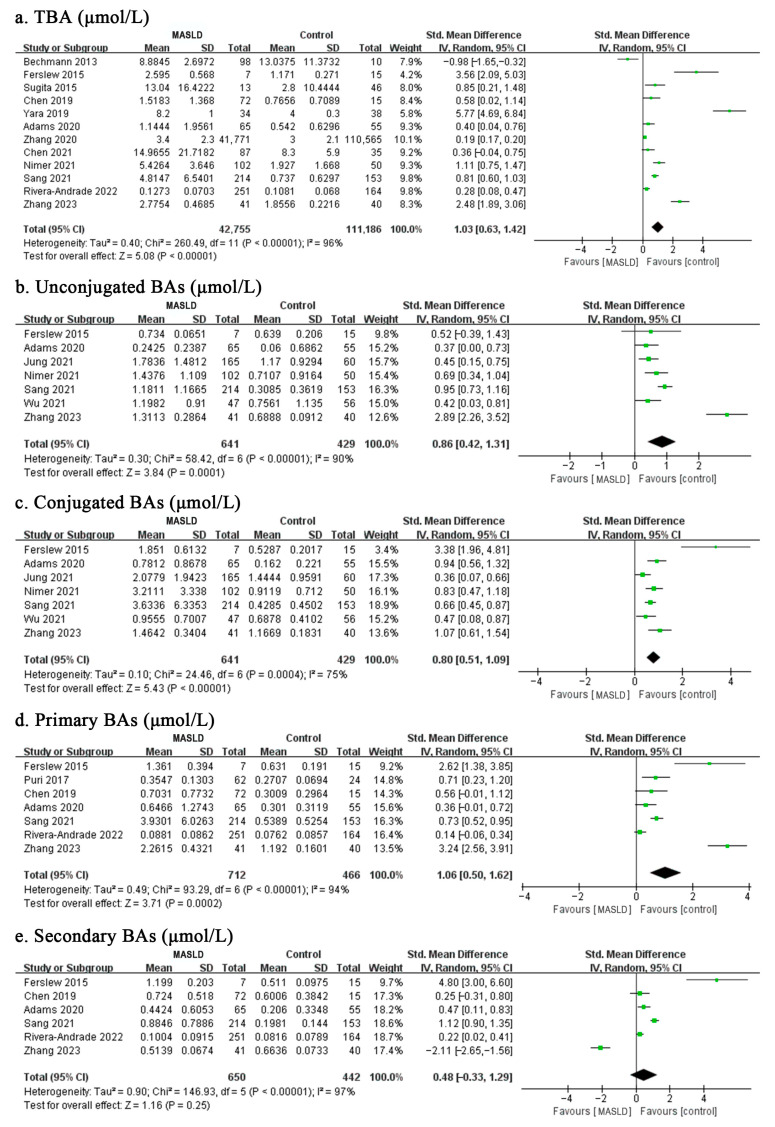
Meta-analysis comparing the concentrations of TBA (**a**), unconjugated BAs (**b**), conjugated BAs (**c**), primary BAs (**d**), and secondary BAs (**e**) between MASLD patients and healthy controls. Abbreviations: MASLD, metabolic dysfunction-associated steatotic liver disease; TBA, total bile acid; BA, bile acid [17,18,19,20,21,22,23,24,25,26,27,28,29,30,31,32,33,34,35].

**Figure 2 biomolecules-13-01356-f002:**
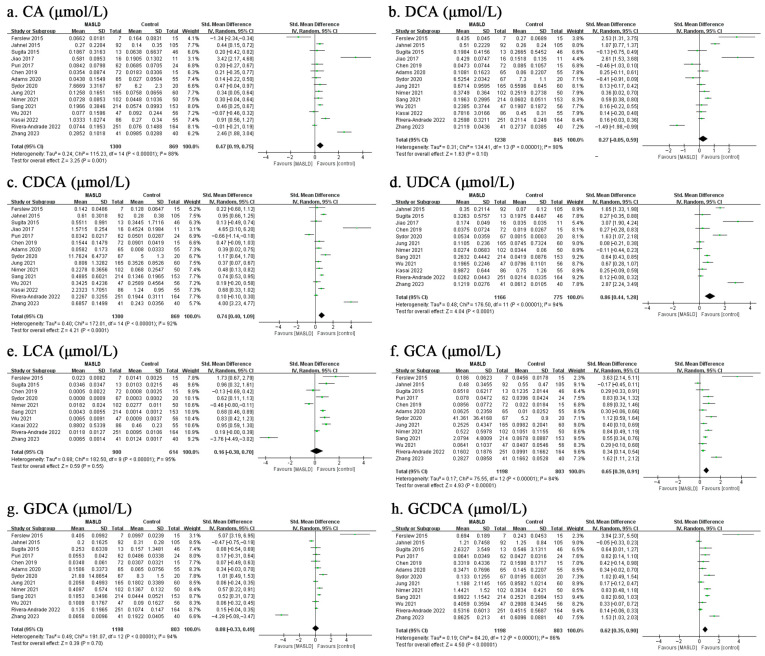
Meta-analysis comparing the concentrations of CA (**a**), DCA (**b**), CDCA (**c**), UDCA (**d**), LCA (**e**), GCA (**f**), GDCA (**g**), and GCDCA (**h**) between MASLD patients and healthy controls. Abbreviations: MASLD, metabolic dysfunction-associated steatotic liver disease; CA, cholic acid; DCA, deoxycholic acid; CDCA, chenodeoxycholic acid; UDCA, ursodeoxycholic acid; LCA, lithocholic acid; GCA, glycocholic acid; GDCA, glycodeoxycholic acid; GCDCA, glycochenodeoxycholic acid [18,19,20,21,22,23,24,25,26,27,28,29,30,31,32,33,34,35].

**Figure 3 biomolecules-13-01356-f003:**
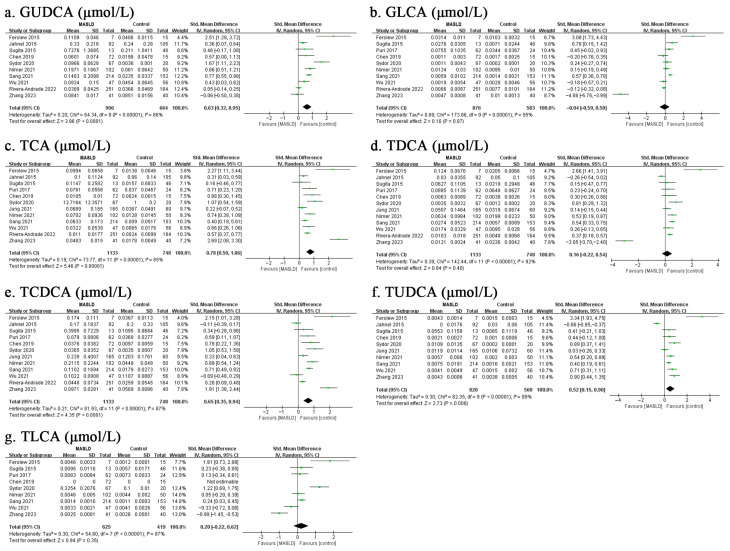
Meta-analysis comparing the concentrations of GUDCA (**a**), GLCA (**b**), TCA (**c**), TDCA (**d**), TCDCA (**e**), TUDCA (**f**), and TLCA (**g**) between MASLD patients and healthy controls. Abbreviations: MASLD, metabolic dysfunction-associated steatotic liver disease; GUDCA, glycoursodeoxycholic acid; GLCA, glycolithocholic acid; TCA, taurococholic acid; TDCA, taurodeoxycholic acid; TCDCA, taurochenodeoxycholic acid; TUDCA, tauroursodeoxycholic acid; TLCA, taurolithocholic acid [18,19,20,21,22,23,24,25,26,27,28,29,30,31,32,33,34,35].

**Table 1 biomolecules-13-01356-t001:** Characteristics of the included studies in the meta-analysis.

Study	Location	Study Design	Study Group	Diagnostic Methods	No. of MASLD/ Control	Age (Years)		Sex (Male/Total)		BMI (kg/m^2^)		ALT (U/L)		Bile Acid Detection Method	Quality Score
						MASLD	Control	MASLD	Control	MASLD	Control	MASLD	Control		
Bechmann 2013 [17]	USA	CCS	Adult	Histology	98/10	43.8 ± 3.3 ^‡^	26.0 ± 7.6	23/98	7/10	52.6 ± 1.7 ^‡^	22.4 ± 2.5	43.2 ± 13.1 ^‡^	19.0 ± 3.8	EA	7
Ferslew 2015 [18]	USA	CCS	Adult	Histology	7/15	48.0 ± 10.0	43.0 ± 12.0	3/7	7/15	32.0 ± 5.2	25.0 ± 2.7	75.0 ± 36.0	33.0 ± 11.0	UPLC−TQMS	8
Jahnel 2015 [19]	Italy, Austria	CCS	Children	Histology	92/105	11.4 ± 5.1 ^‡^	16.0 ± 3.0	51/92	48/105	26.6 ± 3.6 ^‡^	20.5 ± 4.2	79.7 ± 51.9 ^‡^	24.0 ± 28.0	HPLC-MS/MS	8
Sugita 2015 [20]	Japan	CCS	Adult	Histology	13/46	62.5 ± 16.5	20.0~39.0 ^†^	6/13	25/46	25.5 ± 2.8	18.5~25.0 ^†^	64.1 ± 63.6	12.0 ± 12.6	UPLC−TQMS	6
Jiao 2017 [21]	USA	CCS	Children	Histology	16/11	13.7 ± 2.4	12.8 ± 4.2	9/16	6/11	33.8 ± 7.7	19.2 ± 3.4	54.1 ± 29.7	19.4 ± 4.4	LC–MS/MS	8
Puri 2017 [22]	USA	CS	Adult	Histology	62/24	56.6 ± 9.4 ^‡^	39.2 ± 12.4	18/62	11/24	33.7 ± 4.8 ^‡^	27.3 ± 5.8	52.4 ± 27.7 ^‡^	22.7 ± 15.5	LC–MS/MS	7
Chen 2019 [23]	China	CCS	Adult	Histology	72/15	43.0 ± 12.9 ^‡^	40.4 ± 9.4	46/72	9/15	28.1 ± 4.6 ^‡^	21.5 ± 1.5	58.7 ± 49.1 ^‡^	16.0 ± 5.1	UPLC−TQMS	8
Yara 2019 [24]	Japan	CCS	Adult	Histology	34/38	59.6 ± 2.6	56.9 ± 2.3	22/34	21/38	27.5 ± 0.9	22.3 ± 2.2	77.4 ± 14.9	18.8 ± 4.4	LC–MS/MS	8
Adams 2020 [25]	Australia	CCS	Adult	Histology	65/55	49.8 ± 10.3 ^‡^	45.6 ± 10.0	21/65	5/55	41.9 ± 8.9 ^‡^	42.7 ± 8.9	49.7 ± 50.5 ^‡^	30.0 ± 15.0	UPLC−TQMS	8
Sydor 2020 [26]	Germany	CCS	Adult	Ultrasound/ Histology	67/20	61.1 ± 9.8 ^‡^	23.3 ± 2.8	49/67	12/20	30.4 ± 1.8 ^‡^	23.3 ± 2.5	60.3 ± 17.0 ^‡^	25.1 ± 3.3	HPLC–MS/MS	7
Zhang 2020 [27]	China	CS	Adult	Ultrasound	41,771/ 110,565	48 ± 12	44 ± 14	31,510/ 41,771	51,023/ 110,565	27.1 ± 2.8	22.9 ± 2.8	37 ± 28	22 ± 24	EA	8
Chen 2021 [28]	China	CCS	Adult	Histology	87/35	44.4 ± 13.3 ^‡^	33.0 ± 11.8	47/87	27/35	28.2 ± 4.5 ^‡^	23.5 ± 2.7	67.5 ± 41.6 ^‡^	32.4 ± 18.6	LC–MS/MS	8
Jung 2021 [29]	Korea	CCS	Adult	Histology	165/60	55.3 ± 16.7 ^‡^	58.3 ± 13.5	50/107	15/29	27.3 ± 4.3 ^‡^	24.3 ± 2.9	49.0 ± 47.3 ^‡^	24.1 ± 17.3	UPLC−TQMS	7
Nimer 2021 [30]	USA	CCS	Adult	Histology	102/50	52.3 ± 10.6 ^‡^	NR	53/102	NR	32.8 ± 5.8 ^‡^	NR	52.0 ± 48.7 ^‡^	NR	HPLC–MS/MS	6
Sang 2021 [31]	China	CCS	Adult	Histology	153/214	44.8 ± 14.8 ^‡^	38.9 ± 11.4	109/214	108/153	27.8 ± 4.8 ^‡^	22.8 ± 3.0	82.2 ± 60.1 ^‡^	31.5 ± 15.3	UPLC−TQMS	8
Wu 2021 [32]	China	CCS	T2DM patients	Ultrasound	47/56	72.8 ± 6.9 ^‡^	68.3 ± 13.1	37/47	42/56	25.4 ± 1.5 ^‡^	24.1 ± 3.0	23.41 ± 1.8 ^‡^	17.9 ± 7.6	HPLC–MS/MS	7
Kasai 2022 [33]	Japan	CS	Adult	Histology	86/55	59.4 ± 13.0 ^‡^	60.0 ± 15.4	NR	NR	21.2 ± 2.4 ^‡^	28.0 ± 4.2	59.8 ± 38.5 ^‡^	17.5 ± 8.0	LC–MS/MS	7
Rivera-Andrade 2022 [34]	Guatemala	CS	Adult	FLI	251/164	54.3 ± 10.0	56.9 ± 11.3	84/251	82/164	NR	NR	NR	NR	LC–MS/MS	7
Zhang 2023 [35]	China	CS	Adult	Ultrasound/Histology	41/40	39.8 ± 2.2	39.0 ± 2.0	19/41	19/40	28.1 ± 0.5	21.2 ± 0.4	58.4 ± 7.5	14.6 ± 1.0	UPLC−TQMS	6

Abbreviations: MASLD, metabolic dysfunction-associated steatotic liver disease; BMI, body mass index; ALT, alanine aminotransferase; CCS, case-control study; CS, cross-sectional study; UPLC–TQMS, ultra-performance liquid chromatography–triple quadrupole mass spectrometry; HPLC–MS/MS, high-performance liquid chromatography–electrospray tandem mass spectrometry; LC–MS/MS, liquid chromatography with mass spectrometry; EA, enzymatic assay; NR, not reported; T2DM, type 2 diabetes mellitus; FLI, fatty liver index. Values are expressed as mean ± SD. ^†^ means this study only reported the range. ^‡^ means that the data was not recorded directly but could be calculated from the available data according to the following formulas: Total X = (X1 × N1 + X2 × N2)/(N1 + N2); Total SD = ((N1-1) × (SD1) ^2^ + (N2-1) × (SD2) ^2^)/(N1 + N2-2). In these formulas, X, SD, and N refer to the means, standard deviation, and sample size of each group.

**Table 2 biomolecules-13-01356-t002:** Subgroup analysis of circulating bile acid levels in MASLD patients compared with controls by geographic location.

Characteristics	Eastern Countries				Western Countries			
	Study	SMD (95% CI)	*p* ^†^	*I* ^2^	Study	SMD (95% CI)	*p* ^†^	*I* ^2^
Total bile acids	7	**1.40 (0.75, 2.05)**	**<0.001**	97%	5	0.61 (−0.05, 1.28)	0.07	92%
Total unconjugated bile acids	4	**1.12 (0.41, 1.83)**	**0.002**	94%	3	**0.53 (0.29, 0.77)**	**<0.001**	0%
Total conjugated bile acids	4	**0.61 (0.36, 0.86)**	**<0.001**	58%	3	**1.31 (0.57, 2.05)**	**<0.001**	83%
Total primary bile acids	3	**1.49 (0.12, 2.86)**	**0.03**	96%	4	**0.65 (0.12, 1.18)**	**0.02**	84%
Total secondary bile acids	3	−0.24 (−2.14, 1.67)	0.81	98%	3	**1.06 (0.15, 1.98)**	**0.02**	92%
Cholic acid (CA)	7	**0.63 (0.17, 1.08)**	**0.007**	90%	8	0.31 (−0.05, 0.66)	0.09	84%
Deoxycholic acid (DCA)	7	−0.12 (−0.58, 0.33)	0.59	91%	7	**0.69 (0.19, 1.19)**	**0.007**	91%
Chenodeoxycholic acid (CDCA)	7	**0.87 (0.32, 1.42)**	**0.002**	93%	8	**0.64 (0.16, 1.12)**	**0.009**	91%
Ursodeoxycholic acid (UDCA)	7	**0.69 (0.21, 1.17)**	**0.005**	91%	5	**1.16 (0.29, 2.03)**	**0.009**	96%
Lithocholic acid (LCA)	6	−0.04 (−0.96, 0.88)	0.93	97%	4	0.35 (−0.23, 0.93)	0.23	88%
Glycocholic acid (GCA)	6	**0.65 (0.32, 0.98)**	**<0.001**	77%	7	**0.69 (0.27, 1.10)**	**0.001**	88%
Glycodeoxycholic acid (GDCA)	6	−0.51 (−1.33, 0.30)	0.22	96%	7	**0.49 (0.03, 0.95)**	**0.04**	90%
Glycochenodeoxycholic acid (GCDCA)	6	**0.64 (0.27, 1.01)**	**<0.001**	82%	7	**0.63 (0.23, 1.03)**	**0.002**	87%
Glycoursodeoxycholic acid (GUDCA)	5	**0.45 (0.13, 0.77)**	**0.006**	67%	5	**0.91 (0.32, 1.49)**	**0.002**	92%
Glycolithocholic acid (GLCA)	5	−0.72 (−1.94, 0.49)	0.24	97%	5	0.51 (−0.09, 1.11)	0.10	88%
Taurococholic acid (TCA)	6	**0.80 (0.26, 1.34)**	**0.004**	91%	6	**0.72 (0.44, 1.00)**	**<0.001**	69%
Taurodeoxycholic acid (TDCA)	6	−0.24 (−0.97, 0.49)	0.52	95%	6	**0.49 (0.08, 0.90)**	**0.02**	86%
Taurochenodeoxycholic acid (TCDCA)	6	**0.65 (0.20, 1.10)**	**0.005**	88%	6	**0.65 (0.23, 1.07)**	**0.002**	87%
Tauroursodeoxycholic acid (TUDCA)	6	**0.46 (0.21, 0.71)**	**<0.001**	61%	4	0.82 (−0.22, 1.86)	0.12	95%
Taurolithocholic acid (TLCA)	5	−0.21 (−0.78, 0.36)	0.47	88%	4	0.69 (−0.10, 1.39)	0.051	86%

Abbreviations: MASLD, metabolic dysfunction-associated steatotic liver disease; SMD, standardized mean difference; CI, confidence interval. *p* ^†^ denotes the *p*-value for statistical significance based on the Z test. Bold type indicates the SMD (95% CI) and *p* ^†^ of the bile acid species with statistically significant results (*p* < 0.05).

**Table 3 biomolecules-13-01356-t003:** Subgroup analysis of studies comparing the association between circulating bile acid levels and MASLD by disease severity.

Characteristics	Non-MASH vs. Control				MASH vs. Control				MASH vs. Non-MASH			
	Study	SMD (95% CI)	*p* ^†^	*I* ^2^	Study	SMD (95% CI)	*p* ^†^	*I* ^2^	Study	SMD (95% CI)	*p* ^†^	*I* ^2^
Total bile acids	3	0.04 (−1.06, 1.15)	0.94	89%	6	**1.59 (0.53, 2.65)**	**0.003**	96%	3	0.73 (−0.77, 2.23)	0.34	97%
Total unconjugated bile acids	2	**0.42 (0.17, 0.67)**	**0.001**	0%	3	**0.73 (0.39, 1.08)**	**<0.001**	56%	1	0.21 (−0.09, 0.52)	0.17	NR
Total conjugated bile acids	2	0.09 (−0.66, 0.83)	0.82	88%	3	**1.25 (0.53, 1.97)**	**<0.001**	87%	1	**1.28 (0.94, 1.62)**	**<0.001**	NR
Total primary bile acids	2	**0.67 (0.24, 1.11)**	**0.002**	0%	4	**0.92 (0.44, 1.40)**	**<0.001**	69%	2	−0.07 (−0.49, 0.34)	0.73	28%
Total secondary bile acids	1	0.24 (−0.41, 0.89)	0.47	NR	3	**1.54 (0.03, 3.06)**	**0.043**	94%	1	−0.56 (−1.06, −0.06)	0.03	NR
Cholic acid (CA)	4	0.24 (−0.01, 0.48)	0.06	17%	7	0.40 (−0.06, 0.85)	0.09	84%	3	−0.17 (−0.45, 0.11)	0.22	25%
Deoxycholic acid (DCA)	3	−0.33 (−1.14, 0.48)	0.42	90%	6	0.44 (−0.41, 1.28)	0.31	95%	2	−0.04 (−0.30, 0.23)	0.79	0%
Chenodeoxycholic acid (CDCA)	4	**0.26 (0.05, 0.48)**	**0.02**	0%	7	**0.71 (0.11, 1.32)**	**0.02**	91%	3	−0.05 (−0.39, 0.28)	0.75	45%
Ursodeoxycholic acid (UDCA)	3	0.35 (−0.11, 0.80)	0.14	69%	5	**0.96 (0.34, 1.58)**	**0.002**	90%	2	−0.00 (−0.65, 0.64)	0.99	80%
Lithocholic acid (LCA)	2	0.38 (−0.55, 1.32)	0.42	83%	4	**0.61 (0.12, 1.09)**	**0.02**	73%	1	0.00 (−0.49, 0.49)	1.00	NR
Glycocholic acid (GCA)	4	**0.51 (0.06, 0.96)**	**0.03**	72%	6	**0.99 (0.58, 1.40)**	**<0.001**	78%	3	0.44 (−0.10, 0.98)	0.11	78%
Glycodeoxycholic acid (GDCA)	4	−0.02 (−0.23, 0.19)	0.85	0%	6	**0.64 (0.16, 1.12)**	**0.009**	85%	3	0.19 (−0.04, 0.43)	0.10	0%
Glycochenodeoxycholic acid (GCDCA)	4	0.24 (−0.16, 0.63)	0.24	66%	6	**0.87 (0.47, 1.28)**	**<0.001**	78%	3	0.21 (−0.32, 0.75)	0.43	78%
Glycoursodeoxycholic acid (GUDCA)	2	**0.47 (0.14, 0.81)**	**0.006**	0%	4	**1.22 (0.59, 1.86)**	**<0.001**	82%	1	−0.15 (−0.64, 0.34)	0.55	NR
Glycolithocholic acid (GLCA)	3	−0.08 (−0.46, 0.30)	0.68	38%	5	**0.74 (0.08, 1.40)**	**0.03**	84%	2	**0.36 (0.003, 0.71)**	**0.045**	0%
Taurococholic acid (TCA)	4	**0.45 (0.03, 0.88)**	**0.04**	69%	6	**0.83 (0.47, 1.20)**	**<0.001**	74%	3	**0.47 (0.24, 0.71)**	**<0.001**	0%
Taurodeoxycholic acid (TDCA)	4	0.07 (−0.16, 0.29)	0.57	6%	6	**0.62 (0.29, 0.95)**	**<0.001**	68%	3	**0.40 (0.08, 0.71)**	**0.01**	38%
Taurochenodeoxycholic acid (TCDCA)	4	0.19 (−0.32, 0.71)	0.46	80%	6	**0.84 (0.62, 1.06)**	**<0.001**	33%	3	0.56 (−0.14, 1.25)	0.11	86%
Tauroursodeoxycholic acid (TUDCA)	3	0.39 (−0.09, 0.87)	0.11	72%	5	**0.68 (0.18, 1.18)**	**0.008**	84%	2	0.01 (−0.25, 0.27)	0.95	0%
Taurolithocholic acid (TLCA)	3	**−0.38 (−0.70, −0.06)**	**0.02**	0%	5	**0.77 (0.17, 1.38)**	**0.01**	83%	2	**0.53 (0.02, 1.05)**	**0.04**	NR

Abbreviations: MASLD, metabolic dysfunction-associated steatotic liver disease; MASH, metabolic dysfunction-associated steatohepatitis; non-MASH, simple steatosis; SMD, standardized mean difference; CI, confidence interval; NR, not reported. *p* ^†^ denotes the *p*-value for statistical significance based on the Z test. Bold type indicates the SMD (95% CI) and *p* ^†^ of the bile acid species with statistically significant results (*p* < 0.05).

**Table 4 biomolecules-13-01356-t004:** Meta-regression analysis of the circulating bile acid levels and MASLD.

Characteristics	Age (Years)			Male (%)			BMI (Kg/m^2^)			ALT (U/L)		
	Beta (95% CI)	*p*	aR^2^	Beta (95% CI)	*p*	aR^2^	Beta (95% CI)	*p*	aR^2^	Beta (95% CI)	*p*	aR^2^
Total BA	0.082 (−0.091, 0.255)	0.32	0.4%	3.841 (−4.111, 11.794)	0.31	1.12%	−0.926 (−0.246, 0.061)	0.21	8.8%	0.075 (−0.002, 0.152)	0.06	28.2%
Unconjugated BA	−0.048 (−0.125, 0.029)	0.17	23.8%	−0.625 (−7.649, 6.400)	0.83	−22.9%	−0.041 (−0.214, 0.132)	0.57	−15.2%	0.010 (0.041, 0.062)	0.62	−15.7%
Conjugated BA	−0.018 (−0.619, 0.027)	0.37	12.7%	−0.938 (−4.020, 2.144)	0.49	0.19%	0.028 (−0.045, 0.102)	0.38	54.0%	0.011 (−0.020, 0.042)	0.70	−129.2%
Primary BA	−0.109 (−0.293, 0.075)	0.19	21.7%	1.710 (−9.144, 12.565)	0.70	−18.9%	−0.083 (−0.363, 0.197)	0.46	−8.8%	0.019 (−0.108, 0.145)	0.43	−24.5%
Secondary BA	0.151 (−0.383, 0.685)	0.48	−10.8%	−1.281 (−26.628, 24.065)	0.90	−30.9%	−0.061 (−0.667, 0.789)	0.81	−36.2%	0.097 (−0.183, 0.376)	0.35	3.2%
CA	−0.022 (−0.428, 0.384)	0.89	−32.6%	−1.121 (−19.227, 16.984)	0.87	−31.1%	−0.026 (−0.535, 0.482)	0.89	−41.2%	0.080 (−0.098, 0.258)	0.25	20.1%
DCA	−0.026 (−0.059, −0.007)	0.12	15.3%	−0.700 (−5.791, 4.389)	0.77	−12.0%	0.052 (−0.081, 0.184)	0.41	−6.1%	0.017 (−0.028, 0.061)	0.43	−4.6%
CDCA	**−0.045 (−0.086, −0.004)**	**0.03**	25.8%	1.855 (−4.106, 7.816)	0.51	−6.0%	0.007 (−0.167, 0.181)	0.93	−11.4%	0.006 (−0.053, 0.065)	0.83	−10.3%
UDCA	**−0.034 (−0.067, −0.001)**	**0.04**	30.0%	1.321 (−0.488, 7.523)	0.64	−9.8%	0.111 (−0.109, 0.331)	0.28	−0.4%	0.009 (−0.042, 0.060)	0.71	−11.0%
LCA	0.076 (−0.024, 0.175)	0.12	20.85%	1.630 (−7.589, 10.849)	0.69	−12.5%	−0.056 (−0.444, 0.333)	0.74	−13.2%	0.011 (−0.075, 0.097)	0.77	−14.3%
GCA	0.005 (−0.027, 0.037)	0.76	−15.4%	−0.150 (−3.374, 3.074)	0.92	−20.5%	0.020 (−0.094, 0.134)	0.70	−20.3%	0.006 (−0.027, 0.040)	0.68	−25.4%
GDCA	0.026 (−0.055, 0.106)	0.50	−5.16%	−0.046 (−8.081, 7.988)	0.99	−12.1%	0.093 (−0.187, 0.373)	0.48	−6.2%	0.022 (−0.061, 0.106)	0.56	−9.6%
GCDCA	0.002 (−0.035, 0.038)	0.91	−23.3%	0.270 (−3.812, 4.353)	0.88	21.6%	0.022 (−0.088, 0.132)	0.66	−18.0%	−0.002 (−0.046, 0.042)	0.92	−20.9%
GUDCA	0.007 (−0.024, 0.038)	0.62	−15.8%	1.173 (−2.562, 4.908)	0.49	1.0%	0.150 (−0.026, 0.325)	0.08	33.7%	0.007 (−0.025, 0.040)	0.61	−25.9%
GLCA	−0.059 (−0.108, 0.226)	0.84	−4.2%	−1.147 (−13.160, 10.865)	0.83	−15.3%	0.178 (−0.471, 0.827)	0.53	−10.0%	0.031 (−0.088, 0.151)	0.55	−10.7%
TCA	−0.004 (−0.038, 0.029)	0.79	−14.5%	−0.175 (−3.749, 3.399)	0.92	−14.5%	0.080 (−0.116, 0.276)	0.38	−4.2%	0.001 (−0.034, 0.037)	0.94	−15.9%
TDCA	0.020 (−0.033, 0.073)	0.42	−2.9%	0.408 (−5.380, 6.196)	0.88	−12.0%	0.142 (−0.172, 0.455)	0.33	−1.3%	0.011 (−0.047, 0.068)	0.68	−11.9%
TCDCA	−0.001 (−0.031, 0.028)	0.92	−15.0%	−0.693 (−3.774, 2.389)	0.63	−11.4%	0.112 (−0.423, 0.266)	0.14	17.0%	0.010 (−0.020, 0.040)	0.48	−10.5%
TUDCA	0.019 (−0.016, 0.053)	0.25	30.8%	−0.624 (−6.442, 5.195)	0.81	−28.1%	0.162 (−0.082, 0.405)	0.16	8.4%	−0.002 (−0.043, 0.038)	0.91	−23.5%
TLCA	−0.008 (−0.053, 0.036)	0.66	−16.4%	−1.796 (−7.854, 4.262)	0.50	−12.9%	0.181 (−0.034, 0.395)	0.09	32.6%	0.019 (−0.018, 0.057)	0.25	8.1%

Abbreviations: MASLD, metabolic dysfunction-associated steatotic liver disease; BMI, body mass index; ALT, alanine aminotransferase; CI, confidence interval; aR^2^, adjusted R square; BA, bile acid; CA, cholic acid; DCA, deoxycholic acid; CDCA, chenodeoxycholic acid; UDCA, ursodeoxycholic acid; LCA, lithocholic acid; GCA, glycocholic acid; GDCA, glycodeoxycholic acid; GCDCA, glycochenodeoxycholic acid; GUDCA, glycoursodeoxycholic acid; GLCA, glycolithocholic acid; TCA, taurococholic acid; TDCA, taurodeoxycholic acid; TCDCA, taurochenodeoxycholic acid; TUDCA, tauroursodeoxycholic acid; TLCA, taurolithocholic acid. Bold type indicates the Beta (95% CI) and *p* of the bile acid species with statistically significant results (*p* < 0.05).

## Data Availability

Data described in the manuscript, code book, and analytic code will be made publicly and freely available without restriction.

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
