# Peer review of "Alterations in Circulating Bile Acids in Metabolic Dysfunction-Associated Steatotic Liver Disease: A Systematic Review and Meta-Analysis"

_biomolecules, 2023, doi:10.3390/biom13091356_

Round 1
Reviewer 1 Report
In the current manuscript, Lai and Luo et al. conducted a systematic review and meta-analysis to determine the difference in serum bile acid profiles between MASLD and healthy subjects. This is an interesting area since bile acids have been shown to be an essential regulator in metabolic control and are closely linked to NASH. The results and conclusions are relatively clear.
I would like to provide some comments here which hopefully can improve the quality of the manuscript:
1. In the manuscript, the authors used the expression of the serum bile acid profiles. However, some studies, for example, reference 30, used plasma samples for bile acid analysis. Can the authors comment on the difference in plasma and serum bile acid profiles? Can they be mixed together when doing the analysis?
2. It is not surprising to see that a number of bile acids increased in MASLD patients which reflects the liver injury. A more interesting thing is that is there any difference in bile acid profile in patients with different stages of MASLD?
Easy to read.
Author Response
Response to Reviewer 1 Comments
Point 1: In the manuscript, the authors used the expression of the serum bile acid profiles. However, some studies, for example, reference 30, used plasma samples for bile acid analysis. Can the authors comment on the difference in plasma and serum bile acid profiles? Can they be mixed together when doing the analysis?
Response 1: Thank you very much for your positive feedback and interest in our study. We totally agree that there is a difference between serum and plasma samples in bile acid profiles differences. We have already corrected the expression of serum bile acid profile to circulating bile acid profile. And we have supplemented the subgroup analysis to demonstrate that serum or plasma bile acids profiles were similar with the overall results (Table S5, and Page 11, lines 266-269).
Point 2: It is not surprising to see that a number of bile acids increased in MASLD patients which reflects the liver injury. A more interesting thing is that is there any difference in bile acid profile in patients with different stages of MASLD?
Response 2: Thanks for your very enlightening reminding. Based on limited data, we have divided MASLD patients into MASH and non-MASH patients and performed subgroup analysis (Table 3). It was found that non-MASH and MASH exhibited different circulating BA profiles (Table 3). Compared with control populations, a significant increase in total BA levels was found in patients with MASH (SMD=1.59, 95% CI: 0.53-2.65, P=0.003) but not in those with non-MASH (SMD=0.04, 95% CI: -1.06-1.15, P=0.94). A stratified analysis revealed that non-MASH patients had elevated primary and unconjugated BA levels, whereas MASH patients had higher circulating levels of not only primary and unconjugated BAs but also secondary and conjugated BAs. Further-more, non-MASH patients presented increased CDCA, GCA, GUDCA and TCA con-centrations and decreased TLCA levels, with total SMDs of 0.26 (95% CI: 0.05-0.48, P=0.02), 0.51 (95% CI: 0.06-0.96, P=0.03), 0.47 (95% CI: 0.14-0.81, P=0.006), 0.45 (95% CI: 0.03-0.88, P=0.04), and -0.38 (95% CI: -0.70, -0.06, P=0.02), respectively. However, MASH patients had elevated concentrations of almost all BA molecular species except CA and DCA. Notably, the pooled results revealed that total conjugated BA levels were higher in patients with MASH than in those non-MASH (SMD=1.28, 95% CI: 0.94-1.62, P<0.001); more specifically, TLCA, TCA, TDCA and GLCA were the molecular species of BAs that could distinguish between MASH and non-MASH (SMD=0.53, 95% CI: 0.02-1.05, P=0.04 for TLCA; SMD=0.47, 95% CI: 0.24-0.71, P<0.001 for TCA; SMD=0.40, 95% CI: 0.08-0.71, P=0.01 for TDCA; and SMD=0.36, 95% CI: 0.003-0.71, P=0.045 for GLCA).However, there was insufficient data to perform a subgroup analysis based on the stage of liver fibrosis or steatosis.
Reviewer 2 Report
I have studied the manuscript entitled "Alterations of circulating bile acids in metabolic dysfunction-associated steatotic liver disease: a systematic
review and meta-analysis" by Lai J. et al.
The manuscript covers an interesting topic concerning the profile of the bile acids in patients with the newly defined metabolic dysfunction-associated steatotic liver disease (MASLD), formerly referred as nonalcoholic fatty liver disease (NAFLD).
The present study is a systematic review and meta-analysis. The methodological approach is correect, the results are presented in detail, and the manuscript is well prepared. The language used is almost free of grammatical and syntax errors.
In its present form the manuscript is of adequate quality. Hoewever, before considering publication, the authors are wellcome to dicuss the issues referred below in an effort to ameliorate the final result.
Major issue
1. Line 74: In the relevant PROSPERO record (#CRD42022345481), the retrieval time and the search protocol differ in relation with the content of the manuscript. The authors are requested to properly revise the PROSPERO record.
2. Figures 1, 2, and 3 are not included in the manuscipt file.
Minor issues
1. Line 76: Was there any additional search for unpublished data, especially for dissertations or conference abstracts? Please discuss, focusing to what is referred to the Cochrane handbook (https://handbook-5-1.cochrane.org/chapter_10/10_3_2_including_unpublished_studies_in_systematic_reviews.htm) and the fact that a potential publication bias is referred in the "limitations" paragraph of the "Discussion" section (line 370).
2. Line 140: Please define the exact number of publications identified from each database (PubMed, Embase and Web of Science).
3. Figures S3 and S4 present details that are hardly visible; the authors are requested to provide images of better quality and contrast.
4. The authors are wellcome to additionally perform an analysis of heterogeneity to evaluate the potential effect of the diagnostic methods used in each study (histology vs. others).
5. The authors are wellcome to evaluate their findings using the GRADE Quality of Evidence (see 1) https://www.gradeworkinggroup.org/, and 2) Guyatt GH, et al. GRADE Working Group. What is "quality of evidence" and why is it important to clinicians? BMJ. 2008;336:995-8. DOI: 10.1136/bmj.39490.551019.BE. PMID: 18456631).
English language of very good quality; only minor editing is needed.
Author Response
Response to Reviewer 2 Comments
Major issue
Point 1: Line 74: In the relevant PROSPERO record (#CRD42022345481), the retrieval time and the search protocol differ in relation with the content of the manuscript. The authors are requested to properly revise the PROSPERO record.
Response 1: Thank you for your kind reminding. We have re-registered a PROSPERO record (Submitted ID: 457619) to revise the retrieval time and the search protocol. We could not modify the search protocol directly on our previous PROSPERO record (#CRD42022345481) due to the registration rules of PROSPERO (It does not allow the alterations of records which are over the anticipated completion date that set as 01/08/2023 in our original version. ).
Point 2: Figures 1, 2, and 3 are not included in the manuscipt file.
Response 2: We feel very sorry for the errors. As you suggested, we have uploaded the Figure files.
Minor issues
Point 1: Line 76: Was there any additional search for unpublished data, especially for dissertations or conference abstracts? Please discuss, focusing to what is referred to the Cochrane handbook (https://handbook-5-1.cochrane.org/chapter_10/10_3_2_including_unpublished_studies_in_systematic_reviews.htm) and the fact that a potential publication bias is referred in the "limitations" paragraph of the "Discussion" section (line 370).
Response 1: Thank you for your critical suggestions. We have conducted a comprehensive and systematic search to identify most studies, but unpublished data were not included in this meta-analysis according to our inclusion and exclusion criteria. And we have added the limitations as follows:” there was possible publication bias in the current study. When screening the records, unpublished studies such as dissertations or conference abstracts were excluded. However, trim-and-fill analyses indicated that the impact of this bias on our results was likely insignificant, and the sensitivity of the publication bias test was considered low due to the limited number of included studies”. (Page 14, lines 399-403).
Point 2: Line 140: Please define the exact number of publications identified from each database (PubMed, Embase and Web of Science).
Response 2: Thanks for your important suggestion. We have added the exact number of publications identified from each database. (Page 4, lines 144-146 and Modified Figure S1).
Point 3: Figures S3 and S4 present details that are hardly visible; the authors are requested to provide images of better quality and contrast.
Response 3: Thanks for your suggestion. We have already provided images of better quality and contrast. (Modified Figure S3 and S4).
Point 4: The authors are wellcome to additionally perform an analysis of heterogeneity to evaluate the potential effect of the diagnostic methods used in each study (histology vs. others).
Response 4: Thank you for your enlightening suggestion. We have added subgroup analysis based on MASLD diagnostic methods. (Table S6, Page 11, lines 269-271 and Page 14, lines 395-397).
Point 5: The authors are wellcome to evaluate their findings using the GRADE Quality of Evidence (see 1) https://www.gradeworkinggroup.org/, and 2) Guyatt GH, et al. GRADE Working Group. What is "quality of evidence" and why is it important to clinicians? BMJ. 2008;336:995-8. DOI: 10.1136/bmj.39490.551019.BE. PMID: 18456631).
Response 5: Thank you for your suggestion. We have supplemented the GRADE assessment for each outcome of circulating bile acids. (Table S4, Page 3, lines 134-135 and Page 11, lines 283-286).